# Factors Associated with ADL Dependence in Nursing Home Residents with Korsakoff’s Syndrome and Other Alcohol-Related Disorders: An Explorative Cross-Sectional Study

**DOI:** 10.3390/jcm12062181

**Published:** 2023-03-11

**Authors:** Eline S. Böhner, Bea Spek, Karlijn J. Joling, Yvonne Zwaagstra, Ineke J. Gerridzen

**Affiliations:** 1Atlant, Korsakoff Centre of Expertise, Kuiltjesweg 1, 7361 TC Beekbergen, The Netherlands; 2Amsterdam UMC, Epidemiology and Data Science, Meibergdreef 9, 1105 AZ Amsterdam, The Netherlands; 3Department of Medicine for Older People, Amsterdam UMC Location Vrije Universiteit, 1081 HV Amsterdam, The Netherlands; 4Public Health Research Institute, 1081 HV Amsterdam, The Netherlands

**Keywords:** Korsakoff’s syndrome, ADL dependence, activities of daily living, cognitive impairment

## Abstract

Difficulties in performing activities of daily living (ADL) are common in patients with Korsakoff‘s syndrome (KS). The aim of this study was to identify factors associated with ADL dependence in nursing home residents with KS. This exploratory, cross-sectional study included 281 residents with KS from 9 specialized nursing homes in the Netherlands. We examined demographic, cognitive, somatic, and (neuro)psychiatric characteristics. ADL dependence was assessed with the Inter-RAI ADL Hierarchy Scale. Multivariable logistic regression analyses were used to identify factors associated with ADL dependence. Cognitive impairment (odds ratio [OR] = 7.46; 95% confidence interval [CI] = 2.10–30.5), female gender (OR = 3.23; CI, 1.21–8.78), staying in a nursing home for ≥5 years (OR = 3.12; CI, 1.24–8.33), and impaired awareness (OR = 4.25; CI, 1.56–12.32) were significantly associated with higher ADL dependence. Chronic obstructive pulmonary disease (COPD) was significantly associated with lower ADL dependence (OR = 0.31; CI, 0.01–0.84). The model explained 32% of the variance. The results suggest that when choosing interventions aimed at improving ADL functioning, special attention should be paid to residents living more than five years in the nursing home, with a female gender, with more severe cognitive impairments, and/or with COPD.

## 1. Introduction

Korsakoff’s syndrome (KS) is a neuropsychiatric disorder caused by severe thiamine deficiency, predominantly in the context of alcohol abuse and malnutrition [1]. There are no generally accepted criteria and commonly used definitions for diagnosing KS; various terms are used in clinical practice and the literature: “Wernicke Korsakoff syndrome”, “persistent alcohol amnestic disorder”, “Alcohol-induced major neurocognitive disorder, amnestic confabulatory type” [2,3,4]. KS is characterized by severe cognitive deficits, behavioral problems, and various comorbidities such as cardiovascular diseases, chronic obstructive pulmonary disease (COPD), neurological diseases, diabetes mellitus, hypertension, and malignancy [5]. Memory problems and executive function disorders are the most typical cognitive deficits [6]. Further symptoms of KS are apathy, disorders of affect, confabulations, impaired awareness, and social-cognitive problems [1,5,7,8]. About 25% of the patients with KS become dependent on long-term care [9]. In the Netherlands, about 1500 patients with KS reside in specialized nursing homes.

The various physical and cognitive problems in KS have a major impact on the ability to carry out activities of daily living (ADL) [5,10]. ADL is described as “activities that are the same for all people and are carried out daily, because they are elementary and necessary to life [11].

ADL independence leads to a slower decline in mobility and improved quality of life, as well as improved emotional well-being and improved cognitive functioning; finally, encouraging autonomy in ADL tasks can reduce challenging behavior in the elderly and those with dementia [12,13,14,15]. Mood and challenging behavior are in turn important predictors for quality of life [16].

Despite the obvious positive effects of encouraging ADL independence, research also shows that most nursing home residents spend their days inactive and are more dependent in their ADL tasks [17]. This highlights the importance of future interventions that focus on improving the ADL of nursing home residents and adapting these activities to their individual needs [18]. In order to optimize health-related quality of life, cognitive functioning, maintaining mobility, and decreasing challenging behavior, it is important to understand the factors that are most influential on ADL dependence. In this way, patients can be referred to the appropriate type of care [16].

Several studies in elderly care have investigated predictors and factors associated with ADL dependence. Muscle mass, strength, and physical performance were found to be predictors for future ADL dependence [19,20]. Care dependence increased with the worsening of cognitive functions, locomotion impairment, and advanced age, experiencing trouble with pain, taking five or more medications, having a chronic condition, having higher depression scores, and having a lower Mini Mental State Examination (MMSE) score [21]. In nursing home residents with moderate to severe dementia, apathy is found to be most predictive of care dependence [16]. Evidently, findings relating to patients in elderly care and dementia do not automatically extend to nursing home residents with KS, as dementia is known to be a progressive condition, whereas KS is a non-progressive condition [1,2].

As far as we know, no research is performed that focuses on factors that are associated with ADL dependence in residents with KS. This study, therefore, aims to explore a wide range of factors across many domains to examine which factors are associated with ADL dependence in nursing home residents with KS and other alcohol-related disorders.

## 2. Materials and Methods

### 2.1. Study Design

This explorative, cross-sectional observational study used the dataset of the previously conducted exploratory descriptive study called the “Korsakov Study”, collected between 2014 and 2016. The “Korsakoff Study” examined the prevalence and severity of neuropsychiatric symptoms in patients with KS living in Dutch nursing homes [2,5].

### 2.2. Participants

In total, 281 residents with KS from 9 specialized nursing homes, ranging from 7 to 82 participants per nursing home, were included in the initial “Korsakoff-Study”. Due to the heterogeneous nature of the terminology, the term KS is used in this study as an umbrella term to describe the Dutch KS nursing home population in its full scope. All participating nursing homes provide specialized care to KS residents and participate in the “Dutch Korsakoff Knowledge Centre (KKC)”. Details on enrollment, procedures, and results of the study have been previously described [5]. Study inclusion criteria were: (1) a primary diagnosis of KS, Wernicke encephalopathy, Wernicke Korsakoff syndrome, or alcohol-induced persisting amnestic disorder as reported in the medical record; (2) participants had to be admitted to a nursing home for at least three months because of the possibility of a reversible alcohol-related impairment during the first three months of abstinence; and (3) only participants with a legal representative could participate.

### 2.3. Outcome Measure (ADL-H)

ADL dependence was assessed with the Activities of Daily Living Hierarchy Scale of the interRAI-LTCF (ADL-H). This scale measures performance on four self-care tasks (personal hygiene, toilet use, locomotion, and eating) and was found to be reliable and valid for use in home care and mental care [22,23]. The scale has seven levels, indicating the score of each participant. The score can be retrieved from a decision tree: (0) Independent in all four ADLs, (1) supervision in at least one ADL, (2) limited assistance in one or more of the four ADLs, (3) at least extensive assistance in personal hygiene or toilet use, (4) extensive assistance in eating or locomotion, (5) total dependence in eating and/or locomotion, and (6) total dependence in all four ADLs [24]. Two cut-off points on the ADL-H are considered: the same cut-off point is chosen as in the original “Korsakoff Study” in the context of better generalizability: independent or supervised (score 0–1) versus impaired (score 2–6), and one cut-off is considered practically relevant: independent (score 0) versus supervision or impairment (score 1–6). Since “supervision” is a widely used and typical form of ADL support in the population of patients with KS, and also because supervision requires action by the caregiver, we chose to see this as not independent.

### 2.4. Variables

Everyday cognitive performance was measured with the validated InterRAI Cognitive Performance Scale (CPS) [25]. The CPS uses items on decision-making, awareness, memory, expression, and eating performance from the InterRAI. Final scores are calculated using a decision tree: 0 (no cognitive impairment) to 6 (severe cognitive impairment). A CPS score of ≥2 indicates that it is likely that cognitive impairment is present [26].

Awareness of functional deficits was assessed with the Patient Competency Rating Scale (PCRS). The PCRS includes both a self-rating list and an informant rating list. The participant judged his/her own ability to complete several everyday tasks, and this was compared to the ratings of the first responsible nurse or nursing assistant. The discrepancy between the participant’s and nurse’s ratings represents the level of self-awareness [27].

Neuropsychiatric symptoms (NPS) were assessed using the Neuropsychiatric Inventory Questionnaire (NPI-Q). The presence of NPS (yes/no) and their severity (mild = 1, moderate = 2, severe = 3) were determined. The severity of NPS is expressed as a sum score. The NPI-Q is a short questionnaire derived from the more comprehensive Neuropsychiatric Inventory (NPI) [28] and is a valid and reliable instrument to assess NPS in dementia [29,30]. All variables included in the models were chosen based on the previous literature and the clinical expertise of occupational therapists and elderly care physicians working with KS patients. Characteristics and clinical data were collected from the participants’ medical records by the elderly care physicians. The questionnaires, which included the CPS, PCRS, NPS, and NPI-Q, were administered to the primary responsible nurse or nursing assistant and the resident during a structured interview by research interviewers and a research assistant, all trained by the researcher (IG).

### 2.5. Statistical Analysis

We dichotomized the main outcome measure ADL-H according to the degree of ADL dependence at the two cut-off points and used two prespecified full logistic regression models to examine the factors associated with ADL dependence. The “main model” used a cut-off point similar to that used in the original “Korsakoff-Study”, and the “sensitivity analysis model” used a cut-off point similar to that considered clinically relevant. From both models, we reported the outcomes: regression coefficients, confidence intervals, and significance levels and model performance measures: Nagelkerke R^2^ and AIC. For the significant variables (*p* ≤ 0.05) in the models we calculated odds ratios (OR). For the statistical analysis, the package “Stats” in R-Statistics 4.0.4 was used.

### 2.6. Ethical Considerations

All subjects gave their informed consent for inclusion before they participated in the study. The study was conducted in accordance with the Declaration of Helsinki and approved by the Institutional Review Board of the Amsterdam UMC (WC 2014-010 on 30 January 2014) and considered not subject to the Dutch Medical Research Involving Human Subjects Act. Data were stored at Amsterdam UMC and were only accessible through a secured network environment.

## 3. Results

### 3.1. Sociodemographic and Clinical Characteristics

The sociodemographic and clinical characteristics are reported in Table 1. The mean age of the participants was 63.2 years (SD 7.9), and most of them were male (78%). Cardiovascular diseases were the most frequent comorbid somatic disorders (39%), followed by neurologic diseases (28%), and COPD (29%). Mood disorders were the most common psychiatric disorders (31%). The majority of participants used psychotropic drugs (67%), such as antipsychotics, antidepressants, and benzodiazepines. About a quarter of the participants experienced severe cognitive impairment (24%). Most participants (70%) were limited in their awareness of functional deficits, which caused them to overestimate their abilities.

The group distribution according to the initial ADL-H cut-off point in the main model resulted in an unequal distribution of participants in both groups, namely 72 impaired participants (26%) vs. 209 (74%) independent or supervised participants. All characteristics were more prevalent in the ADL-impaired group, except COPD (24% vs. 30%) and personality disorders (10% vs. 11%). The minority of the participants were female (22%) and they were 8% more impaired than males. Lowering the cut-off point on the ADL-H used in the sensitivity analysis increased the size of the ADL-supervised/dependent group: n = 182 (65%), versus the ADL-independent group of n = 99 (35%).

### 3.2. Multivariable Logistic Regression Models

In the main multivariable logistic model, significant associations were found for four variables (Table 2). Participants with severe cognitive impairment (CPS 5–6) showed the strongest association with higher ADL dependence *β* = 2.01, *p* = 0.002, followed by gender *β* = 1.17, *p* = 0.02, and five or more years of residing in a nursing home *β* = 1.13, *p* = 0.02. The presence of COPD showed a negative association with ADL functioning *β* = 1.19, *p* = 0.03. After sensitivity analysis, severe cognitive impairment remained significant: *β* = 1.89, *p* = 0.001. Gender and length of admission were no longer found to be significant. Moderate cognitive impairment *β* = 1.15, *p* = 0.003, and severe impaired awareness *β* = 1.45, *p* = 0.005, were also significantly associated with ADL dependence, contrary to the main model.

OR were calculated for the significant variables of both models (Table 3). The main model shows that the female gender increases the odds of becoming ADL dependent by a factor of 3.23 compared to the male gender. A negative association with an OR of 0.31 was found for COPD. Severe cognitive impairment increases the odds of ADL dependence by a factor of 7.46 compared to participants with mild cognitive impairment. The main model was statistically more robust than the sensitivity analysis model, given the lower AIC (195.5 vs. 242.6).

## 4. Discussion

This study examined demographic and clinical characteristics associated with ADL dependence in nursing home residents with KS and other alcohol-related disorders. The results of this study indicated a strong association between severe cognitive impairment and ADL dependence. Both the main model and the sensitivity analysis indicated that the odds of ADL dependence increase by factors of 7.5 and 6.6, respectively, when the resident is severely cognitively impaired. Length of stay in a nursing home as well as female gender were found to increase the odds of ADL dependence by factors of 3.1 and 3.2, respectively. Surprisingly, COPD was found to be significantly associated with decreased ADL dependence (OR = 0.3). The association with COPD was no longer found to be significant in the sensitivity analysis. However, the sensitivity analysis showed a significant association (OR = 4.3) between level of awareness and ADL dependence. No significant associations were found with age, use of psychotropic drugs, psychiatric disorder, or NPS in both models. Both models explained 32% of the variance in ADL dependence, leaving 68% unexplained.

The strong association between ADL dependence and cognitive impairment we found supports and extends the previous literature reporting similar findings [16,20,31,32,33]. However, these previous studies differed in population and study designs: patients with dementia in all kinds of disease stages and elderly people living in nursing homes were studied in both cross-sectional and longitudinal designs. Nevertheless, cognitive impairment stood out in terms of the strength of its significant association with ADL dependence. A word of caution must be mentioned since most of the previous studies used the Mini Mental State Examination (MMSE) [34] to assess cognitive functioning, whereas in our study the CPS was used.

Surprisingly, age was not found to be significantly associated with ADL dependence, unlike residing more than 5 years in a nursing home. This rather contradictory result suggests that older patients with KS may still be ADL independent, but younger patients who are institutionalized for a longer period may become ADL dependent at some point. It is known that as a result of a long-term stay in a nursing home, institutionalized behavior can occur, such as taking less initiative and thinking less creatively. Residents are increasingly adapting to their environment and becoming less self-reliant [35,36]. Institutionalization has in turn a negative effect on functional mobility, emotional well-being, cognitive functioning, problem behavior, and quality of life [12,13,14,15,37].

In contrast to other studies, which found that ADL dependence is significantly correlated to general and disease-specific health status in patients with advanced COPD [38], we found a negative association between COPD and ADL dependence (OR = 0.3, 95% BI = 1.2–8.3). It should be mentioned that the GOLD classification, which classifies the severity of COPD, was not investigated in our sample. The severity of COPD is therefore unknown; it is presumable that the severity of COPD in our sample was low. In addition, there is a possibility that the COPD patients in the sample were less cognitively impaired; however, this was beyond the scope of this study. Another explanation could be that because of multidisciplinary care programs [39], which focus on the treatment of COPD and its associated functional limitations, there is a strong emphasis on encouraging resident’s independence, which makes patients less dependent. However, it should also be mentioned that it is often underestimated how much help a COPD patient really needs and that there is a discrepancy between what the patient says they need in terms of help and what a proxy (family member, caregiver) estimates that the patient needs [40]. Thereby, impaired awareness and care avoidance, which are common among patients with KS, may have caused that COPD patients with KS do not receive the appropriate amount of support in ADL and therefore were assessed to be less dependent in this study [1].

It is also somewhat surprising that in our study we found a significant association between female gender and increased ADL dependence, since in previous studies regarding factors associated with ADL dependence, “gender” was not found to be significantly associated. Yet these studies were conducted on community-dwelling elderly, patients with dementia, and elderly care [16,20,21,31]. On the other hand, a large cross-national study comparing sex differences in ADL in Europe has found that women are more likely to be ADL dependent than men and that this difference increases with age [41]. Our study did not investigate whether there were gender differences in age, medication use, or number of comorbidities, which could have introduced bias and increased the probability of a type 1 error. Additional studies are needed to better understand the influence of gender and COPD on ADL dependence in patients with KS.

Factors associated with ADL dependence have been studied more extensively in nursing home residents with dementia. These studies reported that apathy is a strong predictor of ADL dependence [16]. Contrary to our expectations, this study did not find a significant association between apathy and ADL dependence, despite the fact that apathy occurs in 50% of the KS population [5]. There are some possible explanations for this discrepancy. First, the discrepancy can possibly be explained by the fact that Henskens et al. [16] examined patients with a more severe stage of dementia. It would be interesting to further investigate whether apathy in severely cognitively impaired patients with KS is associated with ADL. Second, by using the “Empathic Directive Approach” (EDA) [42], which is widely implemented in Dutch KS care, apathy may not lead to increased dependence in ADL. In the EDB, for example, the initiative for daily activities often comes from the supervisors, and by providing structure and predictability in the environment, patients with KS are activated. Third, it is possible that caregivers have become used to the often poor ADL hygiene of the often care-avoiding KS patient and, as a result, underestimate the level of care needed and the presence of apathy [1].

This study used a large study sample from nine specialized nursing homes spread across the Netherlands [5]. The sample included residents with all forms of KS and other alcohol-related cognitive impairments and thus represents the total KS population in Dutch nursing homes. Some limitations need to be mentioned. First, this study was cross-sectional in design, so there might be some over- or underestimation of ADL dependency scores due to fluctuations in ADL performance over time. Second, due to using a dataset from a previous study, we might have missed possible important factors such as muscular strength, physical endurance, or pain, which all could have an association with ADL performance. Third, it is important to note that ADL dependence was assessed with the ADL-H, which is not validated for use in KS. There is a possibility that the ADL-H was not sensitive enough for the target population, which needed relatively little help or only supervision. Therefore, the association between the characteristics and ADL independence may possibly be over- or underestimated. Accordingly, it would be interesting to see what the outcome would be if the ADL-H score had not been dichotomized. This was, however, beyond the scope of our study. There is a possibility that the type of residential facility may have an effect on the level of dependence on ADLs. Our study did not distinguish between the nine nursing homes. A multilevel model to adjust for the type of residential facility could be considered in a follow-up study. Finally, although there was no doubt about meeting the assumption of linearity between age and ADL impairment in the logit transformation from the main model, meeting the assumption in the sensitivity analysis was questionable. To compare both models, we decided not to use a nonlinear term or spline function in this model. The choice for a pre-specified full model limited the use of degrees of freedom, and thus, the risk of overfitting.

Findings of our study may help professional caregivers to better understand the potential risk of becoming ADL-dependent and to choose interventions with the aim to prevent further deterioration in ADL, minimize disability, improve patients’ independence, and delay institutionalization as long as possible [43,44]. Given the findings in this study, it seems important to focus on interventions aimed at improving ADL skills and cognitive functioning. Developing strategies to cope with cognitive limitations may have a beneficial effect on ADL dependence. Accordingly, it seems relevant to assess if COPD is present; KS residents do receive adequate support. In addition, it would be interesting to examine what effect institutionalization has on ADL functioning.

## 5. Conclusions

This nationwide study aimed to identify factors associated with ADL dependence in patients with KS living in nursing homes. Based on the analysis, it can be concluded that several physical and cognitive factors are associated with ADL dependence. The results suggest that when choosing interventions aimed at improving ADL functioning, special attention should be paid to residents living more than five years in the nursing home, with female gender, with more severe cognitive impairments, and/or with COPD. Future research should identify which specific cognitive functions most affect ADL functioning and how functional status and cognitive impairment can be reliably and validly measured in KS. Finally, further research is needed to assess the role of institutionalization and its impact on ADL independence.

## Figures and Tables

**Table 1 jcm-12-02181-t001:** Socio-demographic and clinical characteristics of residents with KS living in nursing homes, main analysis: ADL-H cut-off point independent or supervised (score 0–1)/impaired (score 2–6), sensitivity analysis: ADL-H cut-off point independent (score 0)/supervised or impaired (score 1–6) (*n* = 281).

				Main Model				Sensitivity Analysis			
		Total		ADL Impaired		ADL Independent or Supervision		ADL Impaired		ADL Independent or Supervision	
Characteristic		*n =* 281	(100%)	*n* = 72	(26%)	*n* = 209	(74%)	*n* = 182	(65%)	*n* = 99	(35%)
Age ^1^		63.2	(SD 7.9)	65.5	(SD 8.1)	62.4	(SD 7.7)	63.8	(SD 7.8)	62.1	(SD 8.1)
Gender	Male	219	(78)	52	(72)	167	(80)	144	(79)	75	(76)
	Female	62	(22)	20	(28)	42	(20)	38	(21)	24	(24)
Length of stay in nursing home	Short (<5 years)	138	(49)	27	(38)	111	(53)	81	(45)	57	(58)
	Long (≥5 years)	143	(51)	45	(63)	98	(47)	101	(55)	42	(42)
Use of psycho-tropic drugs		185	(67)	52	(72)	133	(64)	126	(69)	59	(60)
Somatic disorder	Cardio-vascular diseases	114	(39)	38	(53)	76	(36)	79	(43)	35	(35)
	COPD	80	(29)	17	(24)	63	(30)	54	(30)	26	(26)
	Neurologic diseases	69	(28)	22	(31)	47	(22)	48	(26)	21	(21)
Psychiatric disorder	Mood disorder	85	(31)	22	(31)	63	(30)	56	(31)	29	(29)
	Psychotic disorder	48	(17)	16	(22)	32	(15)	38	(21)	10	(10)
	Personality disorder	31	(12)	7	(10)	24	(11)	24	(13)	7	(7)
	Obsessive compulsive disorder ^a^	38	(12)	10	(14)	28	(13)	27	(15)	11	(11)
CPS mean score (range 0–6)	No or mild cognitive impairment (CPS 0–1)	89	(32)	8	(11)	81	(39)	38	(18)	51	(52)
	Moderate cognitive impairment (CPS 2–4)	124	(44)	32	(44)	92	(44)	86	(47)	38	(38)
	Severe cognitive impairment (CPS 5–6)	68	(24)	32	(44)	36	(17)	58	(32)	10	(10)
PCRS awareness discrepancy score ^b^	No or mild impairment (score < 28)	57	(30)	9	(13)	48	(29)	27	(15)	30	(30)
	Moderate impairment (score 28–51)	73	(39)	13	(18)	60	(29)	45	(25)	28	(28)
	Severe impairment (score > 51)	59	(31)	18	(25)	41	(20)	48	(26)	11	(11)
NPI-Q total se-verity score (o-36) ^2^		8	(IQR 8.0)	10	(IQR 8.3)	7	(IQR 7.0)	8.5	(IQR 9)	6	(IQR 6.5)
NPI-Q Symptom	Irritability/lability	192	(68)	53	(74)	139	(67)	126	(69)	66	(67)
	Agitation/aggression	165	(59)	46	(64)	119	(57)	110	(60)	55	(56)
Apathy/indifference severity	Mild	139 56	(51) (20)	44 15	(61) (21)	95 41	(45) (20)	82 38	(45) (21)	60 18	(61) (18)
	Moderate	52	(19)	16	(22)	36	(17)	37	(20)	15	(15)
	Severe	31	(12)	13	(18)	18	(9)	25	(14)	6	(6)

SD—Standard deviation; IQR—interquartile range; CPS—Cognitive Performance Scale; PCRS—Patient Competency Rating Scale; NPI-Q—Neuro Psychiatric Inventory Questionnaire; ^1^ Mean (SD) reported; ^2^ Median (IQR) reported; ^a^ Including hoarding; ^b^
*n* = 189.

**Table 2 jcm-12-02181-t002:** Associations between characteristics and ADL dependence after multivariable logistic regression according to the main model (independent/supervised versus impaired) and sensitivity analysis (independent versus supervised/impaired).

Variable		Main Model		Sensitivity Analysis	
		*β*	*p* Value	*β*	*p* Value
Age		0.02	0.41	−0.01	0.63
Gender	Male	reference	reference	reference	reference
	Female	1.17	0.02 *	−0.01	0.96
Length of stay in nursing home	Short (<5 years)	reference	reference	reference	reference
	Long (≥5 years)	1.13	0.02 *	0.56	0.15
Use of psychotropic drugs		−0.08	0.89	0.002	0.99
Somatic disorder	Cardiovascular diseases	0.01	0.98	0.22	0.58
	COPD	−1.19	0.03 *	−0.19	0.66
	Neurologic diseases	−0.22	0.70	0.28	0.54
Psychiatric disorder	Mood disorder	−0.52	0.31	−0.60	0.16
	Psychotic disorder	0.63	0.24	0.81	0.11
	Personality disorder	−0.69	0.37	0.68	0.26
	Obsessive compulsive disorder ^a^	−0.90	0.23	−0.47	0.44
CPS mean score (range 0–6)	No or mild cognitive impairment (CPS 0–1)	reference	reference	reference	reference
	Moderate cognitive impairment (CPS 2–4)	0.94	0.11	1.15	<0.01 *
	Severe cognitive impairment (CPS 5–6)	2.01	<0.01 *	1.89	<0.01 *
PCRS awareness discrepancy score ^b^	No or mild impairment (score < 28)	reference	reference	reference	reference
	Moderate impairment (score 28–51)	−0.44	0.45	0.50	0.24
	Severe impairment (score > 51)	0.76	0.23	1.45	<0.01 *
	NPI-Q total severity score (o-36)	0.06	0.25	−0.03	0.57
NPI-Q Symptom	Irritability/lability	0.13	0.81	−0.50	0.27
	Agitation/aggression	0.75	0.16	0.43	0.31
Apathy/indifference	No apathy	reference	reference	reference	reference
	Mild	−0.32	0.58	0.70	0.18
	Moderate	−0.13	0.83	0.76	0.19
	Severe	0.21	0.81	1.04	0.24
Nagelkerke *R^2^*		32		32	
AIC		195.45		242.6	

CPS—Cognitive Performance Scale; PCRS—Patient Competency Rating Scale; NPI-Q—Neuro-Psychiatric Inventory Questionnaire; ^a^ including hoarding; ^b^ n = 189; * *p* < 0.05.

**Table 3 jcm-12-02181-t003:** Characteristics significantly associated with ADL dependence: OR of ADL dependence according to the multivariable logistic main model (independent/supervised versus impaired) and the sensitivity analysis (independent versus supervised/impaired).

Characteristic		Main Model OR (95% CI)	Sensitivity Analysis OR (95% CI)
Gender	Male	reference	reference
	Female	3.23 (1.21–8.78) *	0.99 (0.42–2.34)
Length of stay in nursing home	Short (<5 years)	reference	reference
	Long (≥5 years)	3.12 (1.24–8.33) *	1.75 (0.82–3.85)
COPD		0.31 (0.01–0.84) *	0.83 (0.37–1.89)
CPS	No or mild cognitive impairment (CPS 0–1)	reference	reference
	Moderate cognitive impairment (CPS 2–4)	2.56 (0.12–8.81)	3.16 (1.48–6.94) *
	Severe cognitive impairment (CPS 5–6)	7.46 (2.10–30.5) *	6.64 (2.20–23.23) *
PCRS discrepancy score ^a^	No or mild impairment (score < 28)	reference	reference
	Moderate impairment (score 28–51)	0.65 (0.49–2.04)	1.65 (0.72–3.83)
	Severe impairment (score > 51)	0.75 (0.16–7.50)	4.25, (1.56–12.32) *

OR—Odd’s Ratio; CI—Confidence Interval; CPS—Cognitive Performance Scale; PCRS—Patient Competency Rating Scale; ^a^ n = 189; * *p* < 0.05.

## Data Availability

Data supporting results are saved at the server of the VU University Amsterdam. Due to patient confidentiality, raw data are not made publicly. Anonymized data are available from the authors upon request.

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
