# Peer review of "Factors Associated with ADL Dependence in Nursing Home Residents with Korsakoff’s Syndrome and Other Alcohol-Related Disorders: An Explorative Cross-Sectional Study"

_jcm, 2023, doi:10.3390/jcm12062181_

Round 1

Reviewer 1 Report

From my perspective, this is a good study, well written, with a very interesting sample, a good analysis, and which deals with the fascinating topic of KS. 

Some critical points:

Before using the term "COPD" the term "chronic obstructive pulmonary disease" should be provided, both in the abstract and in the main manuscript (the explanation for COPD is never provided, and some readers will requiere this explanation). Besides this minor point, the inverse relationship between COPD and ADL dependence is surprising and it doesn't appear to make sense from a biological or psychological point of view; however, the explanation that appears in the discussion is of good quality. Is there any possibility that patients with COPD had less severe forms of cognitive impairment? 

Regarding the relationship between female gender and ADL dependence, I think the discussion could be extended as it is not clear why this relationship should be taken as valid. Is it merely a type I error?

"No significant associations were found with age, use of psychotropic drugs, psychiatric disorder or NPS in 231 both models." If the authors could provide more information regarding the psychotropic drugs that were being used, this would be helpful as many academics, clinicians and people in the public believe that this could be a factor that contributes to disability.

Reviewer 2 Report

Methods:  I think the full spelling of KORSAKOFF is needed.

Methods:  2.1 and 2.2 are duplicated in some parts of the description, so please describe them in an organized manner.

Methods: Who evaluated the CPS, PCRS, NPS, and NPI-Q?

Methods:   Please state the significance level.

Results:  The description in 3.1 should be in the methods section. In addition, at least the gender ratio and average age should be stated.

Results:   Did you not throw in the facility variable as an adjustment factor?
